# The relationship of work engagement with job experience, marital status and having children among flexible workers after the Covid-19 pandemic

**Murat Çemberci**[1], **Mustafa Emre Civelek**[2], **Adnan Veysel Ertemel** [3]*, **Perlin Naz Cömert**[1]

**1** Business Administration, Yıldız Technical University, Istanbul, Turkey, **2** Business Administration, Istanbul Commerce University, Istanbul, Turkey, **3** Management Engineering, Istanbul Technical University, Istanbul, Turkey

* ertemelav@itu.edu.tr

**Citation:** Çemberci M, Civelek ME, Ertemel AV, Cömert PN (2022) The relationship of work engagement with job experience, marital status and having children among flexible workers after the Covid-19 pandemic. PLoS ONE 17(11): e0276784. https://doi.org/10.1371/journal.pone.0276784

**Data Availability Statement:** All relevant data are available at: https://doi.org/10.6084/m9.figshare.19368077.v1.

## Abstract

The COVID-19 pandemic has brought about serious consequences in business world practices. Among these, flexible working policies have increased to a great extent. This has resulted in serious problems in the work-life balance. In this context, conditions such as having children and marital status have been important factors that can affect work engagement among flexible workers in the post pandemic era. Therefore, this study investigates the relationship of marital status, job experience and having children with work engagement among white-collar workers who work in flexible hours. Data is collected through surveys from 199 flexible working employees. ANOVA and T-tests were employed to analyze the data. The results indicate that only one of the sub-dimensions of work engagement–namely absorption- changes according to their marital status, and yet, the work engagement is not related to having children. In addition, it is seen that there is a significant relationship between job experience and work engagement.

## Introduction

In the 21st century work environment, most employees need to decide on their own work schedules and workplaces in order to adapt to more individualized work times and family constraints [1]. Therefore, flexible work arrangements can become a solution to reduce the problems of employees by helping them about maintaining a balance between work life and family life [1].

Although the prevalence of flexible working arrangements has increased sharply over the past decade, there has been a dramatic increase in this trend after the outbreak of the COVID-19 pandemic [2]. World Health Organization's (WHO) declaration of the COVID-19 outbreak as an international pandemic on March 11, 2020, have resulted in quarantine and "work from home" practices in many countries [3]. With 603,711,760 global cases and 6,484,136 global deaths in total, Covid-19 stimulated high anxiety and great transformations all over the World

**Funding:** The author(s) received no specific funding for this work.

**Competing interests:** The authors have declared that no competing interests exist.

[4]. Thus, the COVID -19 epidemic caused a radical change in the business world in a short time. Actually, along with closures and transitions to new business regulations, the pandemic has created a significant impact on society and the global economy at large. Quarantine has forced most office workers to work from home and organizations to adapt quickly to the new situation with technological and managerial support and adopt new working practices [5]. During this period, the conflict between work and family responsibilities for families with children has increased. In particular, the work-life balance for many workers has been affected seriously as the boundaries between the responsibilities of work and family life have been blurred [6]. At the same time, COVID -19 has increased the burden on women. Since the pandemic began, 60% of women and 54% of men have stated that their domestic work has increased [7]. Specifically, the pandemic has increased the need for data-driven, short-term, yet sustainable solutions to many problems that arise as a result of a diverse and multi-generational workforce mix with different expectations regarding flexible work arrangements [6]. In their study, Tavares et al., [3] have found out that getting used to working from home is regarded as easy for employees. The same study also indicates that lack of interaction with co-workers, lack of resources related to support infrastructures such as the internet or printer, and compensating for remote work with housework and childcare were highlighted among the main difficulties encountered. Transformations in the world and the working environment require further exploration of flexible work arrangements and examining the extent to which flexible work arrangements can help organizations achieve desired results. Accordingly, flexible working arrangements such as working flexible hours, job sharing and working from home are attracting the attention of organizations. In essence, flexible work arrangements aim at improving employee well-being, work-life balance and firm performance [8].

In 2020, about 56% of workers in the United States had a job that could be done from home at least partially because of the fact that their job was knowledge-based and had no physical work requirements [9]. In a study conducted in 2021, it was seen that 50% of the contract workers preferred to work independently because of the freedom and flexibility it provides [10]. Also, the majority of millennials who work say they prefer flexible work arrangements, including the ability to work from home [11]. According to the results of the 'The Impact of the Coronavirus Outbreak on Business Life Survey' conducted in Turkey with the participation of 167 companies, 103 of which are global and 64 of which are local, the rate of companies that had the practice of working from home before the coronavirus was 45 percent, while it has reached to 95 percent for the head office employees [12]. More than 40,7 percent of the companies had stated that they had difficulties in employee motivation during the pandemic. Companies have accelerated their infra-structural actions to adapt to new working practices and they still continue to take the necessary steps for the adaptation of their employees. More than 72 percent of companies' state that they encourage their employees to use digital solutions in order to accelerate adaptation to remote working [12].

Various studies in the extant literature emphasize the advantages of flexible working, such as increasing productivity, work satisfaction, work-life balance, commitment, reducing absenteeism and work-life conflict [6, 13–15]. Recently, Chua, Myeda ve Teo [16] emphasized that flexible work arrangements is the greatest result of Covid-19 in the workplace even if it presents threats alongside opportunities. Bontrager, Clinton ve Tyner [17] argue that flexible working introduce an opportunity to balance work and life, to decrease turnover, to expedite employee development [17]. Flexible work arrangements are distinguished as one of the 6 elements that contribute to the adaptation of employees to the pandemic [18]. Weideman and Hofmeyr [19] found a positive relationship between flexible working arrangements and work engagement. The study undertaken by Richman et al., [20] also showed that workplace flexibility can increase work engagement. Likewise, Zaman and Ansari [21], who has found a positive

relationship between workplace flexibility and work engagement underline that flexible business alternatives are an important factor affecting work engagement in the IT industry. Sonnentag et al., [22] observed that being psychologically disconnected from work during non-working times is a key factor in the formation of high levels of work engagement. According to Pocock's [23] study, one-third of the employees need but do not demand flexibility in order to achieve better harmony and balance in their work and non-work lives. Some studies have shown that flexible working arrangements are associated with better employee well-being [24–26], higher productivity and performance [8, 13, 27, 28]. In addition to its advantages, flexible and remote working could also lead to some negative consequences. For example, it can be said that working remotely during the pandemic period reduces the effects on work-life balance and increases work-family conflict [6]. According to Palumbo's [29] research, working from home has caused public sector workers to be exposed to increased work-life conflict. In the same study, it was observed that working from home increased work-related fatigue and worsened the perception of work-life balance [29]. In addition, it is seen that flexible working has some disadvantages such as difficulties in career management and performance evaluation as well as difficulty in participating in social activities. Moreover, despite past research and theories emphasizing the relationship between flexibility and work engagement, Timms et al., [30] found a negative relationship between the use of flexible work arrangements and work engagement. As a consequence, it is observed that there are different research results on the effects of flexible working on employees and organizational results. In this context, it seems that past research results with conflicting results and the changes caused by the COVID-19 epidemic underline the need for further research on this issue.

In addition, there is a need for more research on how work engagement, which is an important variable found to be associated with performance, organizational citizenship behavior and customer satisfaction, changes with the spread of flexible working arrangements in the post pandemic era [31]. Many governments dictate severe measures related to the pandemic, which could have psychosocial and mental health repercussions for the population [32]. In these uncertain and extraordinary circumstances, work engagement is a concept that can be helpful in battling the sensitive situation at the workplace.

Specifically, demographic characteristics of the employees are examined with regard to work engagement in this study. This is due to the fact that the personal obligations of the employees and the conflict of their work lives are felt more intensively in the pandemic period. In this context, it is assumed that the level of work engagement among flexible workers will change according to having children, marital status and experience. Although some previous studies show the relationship between demographic variables and work engagement, some other studies show the opposite. Hence, this relationship needs to be elaborated further in the post pandemic period. Having a child and marital status stand out as important factors that may be related to work engagement, especially in the post pandemic era. Smith and Dumas [33], Fukuzaki et al., [34], Poulsen et al., [35] found that the number of children is a factor that has an impact on work engagement. Based on the previous studies, it is assumed that there is a significant relationship between work engagement and having a child in this study. Likewise, the extant literature demonstrates that work engagement will vary according to marital status and job experience [36–38]. Based on the previous studies, this study assumes that there is a relationship between marital status and work engagement in flexible workers. Most past studies show that there is a relationship between job experience and work engagement [39–41]. For this reason, the relationship between job experience and work engagement is also investigated in this study. There is a necessity of re-examining the relationships between the aforementioned variables in the new working environment. In addition, conflicting results in similar studies conducted in the past on this subject underline the need for further research in

this area. Mäkikangas et al., [42] have argued that, while work engagement is a more desirable condition in flexible working since it is associated with energy, effort and passion, it has been understudied among flexible workers. Therefore, in the present study which is conducted in flexible work arrangements, work engagement is particularly scrutinized.

Social exchange and commitment-trust theories constitute the theoretical basis of the study. Social exchange theory states that employees are ready to offer their labor and loyalty in return for the economic and social benefits offered by the organization in the employee-employer relationship [43]. Social exchange theory proposes that flexible work arrangements will reciprocate the employer's effort by generating a sense of gratitude in employees, demonstrating higher motivation and loyalty for their work [44]. The commitment-trust theory argues that when commitment and trust exist together, there will be increased efficiency, productivity and efficiency in the work environment [43]. Furthermore, social role theory suggests that stereotyped beliefs about gender roles are originated from seeing people in different social roles [45]. For instance, women's perceived traits are linked with stereotypes such as nurturing and caregiver roles. In this context, flexible working women might take on additional domestic work because of these traditional stereotyped gender roles which may lead to a decrease in work engagement [45].

In this context, these theories are used as the baseline while investigating whether the engagement of employees who use flexible working options in marital status, level of job experience and whether they have children have changed. The study consists of six parts. After the introduction, a literature review on the conceptual background of the research variables is presented. In the third part, the related concepts are examined with the support of the previous literature and hypotheses are developed accordingly. The fourth section describes the research methodology. In the fifth section, discussion and implications, the results are outlined in relation to the previous research and how research findings can contribute to managers is explained. Finally, the limitations of the research are mentioned in the last part.

## Conceptual background

### Flexible work options

The number of employees in flexible employment options has been on the rise in recent years. Flexible working is defined as the ability to decide where, when and how to work [46]. Among flexible working types are flexitime, in which employees determine the start and end time of work, compressed work week, which includes employees working four long days instead of working five regular days, telecommuting, which refers to employees working from home by making use of information technologies, and part-time work [31]. Spreitzer et al., [47] divided flexible work arrangements into three groups, which are flexibility in work schedule (the ability of the employee to set their own working hours, including compressed work weeks, non-standard working hours such as nights and weekends), flexibility in the location of the work (such as working from home), and flexibility in employment relationships (such as short-term contract workers, freelancers). According to a study conducted on employees participating in flexible working at an International Business Machines Corporation (IBM) company in the USA, 74% of the employees stated that they would leave IBM if it were not for flexible working, and 59% stated that they would leave IBM to find a more flexible work [48]. Abbott et al., [49] found that ignoring the commitment and responsibilities of employees in their personal lives at work has costly organizational consequences such as high absenteeism and increased staff turnover. Accordingly, flexible working arrangements are designed to balance the conflicting personal and work demands of employees [50]. In this context, practices that improve employees' work-life balance are considered to be important since they will also be beneficial to

organizations. Flexible working is recommended as a family-friendly policy that allows balancing work-life responsibilities [48].

There are several advantages and disadvantages to adopting flexible working. Increasing the ability to recruit and retain employees is one of these advantages [6]. New forms of work, such as flexible working, have been shown to increase work engagement and work satisfaction and reduce turnover [20, 46, 51]. Kelliher and Anderson's study [52] showed that there is a strong and positive relationship between flexible working and perceived work quality. In addition, informal teleworking arrangements have been found to have a positive indirect effect on employee performance through organizational commitment and work satisfaction [53]. In addition, it has been observed that flexible working time increases employees' autonomy and feedback perception, increasing productivity and work satisfaction and reducing absenteeism [13]. It has been observed that flexible work arrangements reduce employee stress and increase work-life balance [6]. Similarly, according to Golden et al., [14] study, remote working reduces work-life conflict by enabling employees to better meet family needs. The increase in employees' control over their work schedules leads to organizational support, increased perception of work-family balance and organizational commitment [15, 54]. Flexible working has also enabled older adults who do not want to work full time but have valuable knowledge and job experience to offer to organizations [55]. It has been observed that flexible working increases work engagement and reduces work-family conflict [56]. According to Carless and Wintle [57], flexible solutions such as flexible hours and remote working make organizations more attractive to potential employees. Flexibility has been shown to reduce employee turnover and absenteeism and also to increase productivity [58]. It has been observed that women working in companies that offer flexible work options have higher work satisfaction, lower absenteeism, work more in their own time and work until later in their pregnancy [59]. Scandura and Lankau [60] suggested that as a result of providing flexible work arrangements to employees, employees feel the need to respond to this situation with higher commitment, loyalty and work performance mentioning the positive results of the psychological contract between the employer and the employee. This is particularly common in women. According to commitment and trust theory, when commitment and trust exist together, there is an increase in productivity, productivity and efficiency. Therefore, policies that help reduce the problems of employees and provide work-life balance will increase the employee's commitment to the organization, and the relationship of loyalty and trust between the employee and the employer will also increase. This will also affect organizational performance in a positive way. In a past study, employee satisfaction and work-family balance improved when employers provided flexibility of work schedules [1]. It is claimed that working from home as a flexible working schedule increases employees' control over the spatio-temporal work context and improves the quality of organizational activities [61]. On the other hand, Shifrin and Michel [2] demonstrate that flexible work arrangements result in better physical health and reduced absenteeism, which underlines the positive impact of flexible working on the health of workers. A previous study found that flexible time workers have higher work satisfaction [62]. It has been seen that flexible working policies will increase the well-being of employees by providing higher temporal flexibility [63].

In contrast to the positive results listed above, Timms et al.'s study [30] found a negative relationship between work engagement and flexible work arrangements. Although organizations aim to facilitate the work of employees by providing flexible working arrangements, some studies suggest that managers could see the need for flexibility in employees as a sign of decreasing work engagement [50, 64]. In this context, employees are hesitant to take advantage of the convenience of flexible working, considering that flexible working policies may result in career barriers. Accordingly, employees think that being physically present at the workplace

can better reflect their commitment to work in the eyes of managers compared to flexible working [63]. In addition to these, flexible working could bring about difficulties in managing career development and professional image, declining income, difficulty in participating in social activities at work and professional networking. In addition, some managerial barriers may be encountered in flexible working such as work coordination and scheduling problems, difficulties in evaluating performance and equality issues. As a consequence, both positive and negative effects of flexible working are encountered in the extant literature. This stems from conflicting research results on this subject. At the same time, it has been observed that the attitude towards flexible working options varies depending on gender, sector and previous participation in flexible working [48]. After the Covid-19 pandemic, research related to flexible working has increased and gained importance [32, 65]. Especially, studies concerning the additional burden of women in housework and childcare is frequently discussed [66]. In the aftermath of Covid-19, concerns have begun to emerge that working from home will bring back traditional gender roles [66]. Therefore, gender equality seems to be a prominent theme in flexible work researches after Covid-19 [66].

## Work engagement

Work engagement refers to a positive, emotional-motivational high energy state with a high level of dedication and strong focus on work [67]. It has been shown that work engagement can improve employee performance, health and well-being [68]. Work engagement is defined as a positive work-related mental state characterized by three dimensions: vigor, dedication, and absorption [69]. According to Salanova & Schaufeli [70], work engagement is an indicator of the employees' intrinsic motivation. It has also been found that work engagement is associated with a high level of creativity, task performance, organizational citizenship behavior and customer satisfaction [71]. Accordingly, individuals who are engaged about work are highly energetic, enthusiastic and fully immersed in their business activities. Employees who are engaged find the work energizing and feel a personal satisfaction about the work they do [30]. Understanding the antecedents and consequences of work engagement is important for organizations because lack of work engagement is costly [72].

Vigor is the first dimension of work engagement, which expresses the feeling of high energy and mental resilience while working, putting effort into one's work and being persistent in the face of difficulties in the work. The second dimension of work engagement, which is dedication refers to a strong involvement in one's own work and feeling of meaning, enthusiasm, pride, inspiration and struggle related to one's work. The third dimension is absorption which means that the individual concentrates completely on his work, feels that time passes quickly while doing work, and has difficulty in disconnecting from work [73]. In short, vigor indicates endurance as well as the employee's motivation to exert effort; dedication reflects a strong identification with one's job, as it expresses work engagement and enthusiasm for work, whereas absorption refers to the focus and immersion of employees in the work [74]. According to Kahn [75], employees are more engaged in work situations that offer them more psychological meaning and confidence, and when they are psychologically available, employees tend to me more engaged. The past studies have shown that social support, performance feedback, skill variety, autonomy and learning opportunities from managers and colleagues are positively associated with work engagement [71]. In addition, the extant literature indicates that work engagement is strongly related with financial returns, work performance, in-tention to leave and employee creativity [76–79]. Among the factors that lead to work engagement, one of the most interesting factors is job resources. It refers to physical, social or organizational resources that contribute to reducing business costs, achieving business goals and individual

development. Social support, work control, participation in decision making, feedback, rewards, work security and task diversity can be classified as some of the business resource types. According to previous studies, social support, performance feedback and executive coaching were found to be associated with work engagement [78]. Another study has discovered that job resources such as employee empowerment, managerial support, feedback and development opportunities, and individual resources such as self-efficacy and optimism are among the predictors of work engagement [80]. The concept of work engagement was first theorized by Kahn [75] and then operationalized by Maslach et al., [81]. The present study was guided by Schaufeli and Bakker's conceptualization of work engagement.

**Marital status.** Obviously, employees who have a spouse and/or children possess more life duties than their single counterparts [82]. Managing multiple roles has always been a struggle [83]. Carrying lots of nonwork roles and responsabilities lead to a substantial conflict between work and nonwork roles which awakens the necessity of flexibility at work [82]. Therefore, after Covid-19 among flexible workers marital status stands out as a significant variable. It has been seen that married and single women (either widow or divorced) faced more impediments in their career progress [84].

**Having children.** Children are identified with substantial caregiving demands and work-family pressure [85]. Working parents confront with a lot of problems such as insufficient child care and deficiency of time to effectuate both work and family duties [85]. Previous results demonstrated that, workplace features about flexibility seems to have a powerful effect on working parents' stress and well-being [86]. For this reason, it is essential to investigate the relationship between having children and work engagement among flexible workers after Covid-19.

**Job experience.** Job experience can be defined as extent of experience in a given occupation [87]. McDaniel et al., [87] found a positive correlation between job experience ve job performance. It has been demonstrated before that work engagement level changes according to job experience [39, 40]. Previously, higher engagement levels were found in more experienced employees [88]. Likewise, older age, being married and possessing a bachelor's degree or higher were found to be demonstrative of greater work engagement [36].

## Development of hypotheses

### The relationship between having children and work engagement

Although various past studies have found a relationship between having children and the number of children and work engagement, it is important to re-examine this relationship in the post pandemic world [33–35]. Furthermore, social role theory suggests that stereotyped beliefs about gender roles are originated from seeing people in various roles [45]. For instance, women's characteristics are shaped with stereotypes such as nurturing and caregiver roles. In this context, flexible working women might take on additional domestic work because of these traditional stereotyped gender roles which may lead to a decrease in work engagement [45]. As supported by social role theory, stereotyped roles such as caregiver and nurturing are traditionally associated with women [45]. Therefore, flexible working women may feel the burden of traditional stereotypical beliefs. Study conducted by Engelke et al., [89] which analyzes positive and negative circumstances experienced by families during the Covid-19 pandemic, reveals that the most essential distress is the burdensome balance between work and family life. According to a recent study by Del Boca et al., [90], it has been observed that women have been affected more in terms of increased housework and childcare during the COVID-19 period. The same study indicated child-care work was shared more equally among partners compared to housework. The study of Del Boca et al., [90] showed that working women with

children aged 0–5 have more difficulty in maintaining work-life balance during the COVID-19 period. In another study, the time of work, marriage, child development and family stress have been found to be important [1]. Studies have shown that these flexible working arrangements allow mothers to maintain work hours and income after giving birth [91]. The study of Rajesh and Ekambaram [92], in IT sector, has shown that pregnancy, childbirth, marriage and childcare are among the most significant barriers in terms of career development. In the same study, flexible working arrangements were found to be an important facilitator for women to continue in their careers. At the same time, flexible working has also been shown to have beneficial effects on the mental health and well-being of fathers and children as well as improvements in family and couple relationships [61]. In contrast, a biological study of eleven key indicators of chronic stress levels in the UK found that working mothers of two children were 40% more stressed than the average person in pre-pandemic conditions [93]. In another study, it was found that young individuals have more difficulty in breaking away from work psychologically, but women with children may not have that much difficulty in breaking away from work because they are more involved in housework and child care [94]. In the study of Smith and Dumas [33], single and childless workers were found to be less engaged than those with a family. According to a study conducted among nurses, it was observed that the children of nurses under the age of 6 had a beneficial effect on mental health [95]. It has been observed that living with children has a moderator effect that weakens the relationship between employees' positive work thoughts and their work engagement [96]. In addition, it has been observed that women job experiences more workload, stress and conflict than men, and this increases significantly with the increase in the number of children [97]. In the study of Ten Brummelhuis and Bakker [98], it was found that housework reduces vigor because it reduces relaxation. Therefore, these activities were found to significantly affect the next day's work engagement. According to the study of Fukuzaki et al., [34], there is a significant relationship between the number of children and work engagement. In the research conducted on health workers to investigate the issue of work engagement, employee and manager support, it has been found that having 16 years of or longer job experience, being directly involved in patient care, having children and not working in shifts are positively related to work engagement. In addition, high levels of work engagement were reported by therapists in conditions of low psychological disconnection from work, high income satisfaction, graduate qualifications, high frequency of laughing and having children [99]. According to the study of Chung and Van der Horst [100], flexible working arrangements enable women to continue working after the birth of the first child. It has been observed that women who use flexible working hours are less likely to reduce their working hours after childbirth [100]. Therefore, it has been discovered that flexible working is a tool that improves and preserves the working capacity of individuals in the face of increasing family demands [100]. According to the suggestion of Golden [101], it is suggested that married employees and employees with young children participate in more flexible work. Seemingly, previous studies emphasize that women are overwhelmed by the workload during the Covid-19 pandemic and have difficulty in establishing the balance between work and life [1, 90]. Furthermore, both positive and negative effects of having a child in the flexible working process brought by the pandemic process have been shown in prior studies. In this context, it is essential whether having children among flexible workers during the Covid-19 process is related to employee engagement. In this study, it is assumed that there is a significant relationship between having children and work engagement among flexible workers, with the following hypothesis based on the previous studies.

H1a: There is a significant relationship between having children and dedication among flexible workers

H1b: There is a significant relationship between having children and absorption among flexible workers

H1c: There is a significant relationship between having children and vigor among flexible workers

## The relationship between marital status and work engagement

In a study conducted by Sharma et al., [102], it was discovered that there was no difference in work engagement in terms of gender, marital status and income. Another study undertaken by Shukla, Adhikari and Singh [38], has found a significant difference in the levels of work engagement according to gender, marital status and experience. It has been observed that the level of work engagement is higher in married workers. In another study, marital status was found to have a significant and independent effect on nurses' dedication and total work engagement [37]. This can be explained by the fact that married nurses have more emotional stability than single nurses. Chan et al., [36] study revealed that age, marital status and education level affect work engagement. On the other hand, according to Köse's [103] study, marital status, branch and educational status are not significant factors for teachers' work engagement. Similarly, in Willmer, Westerberg Jacobson, and Lindberg's [104] study, no significant relationship was found between education, marital status, age and work engagement. In addition, Othman and Nasurdin [105] argue that age, marital status and education status are not related to work engagement. Zhang et al., [65] and Balay-Odao et al., [106] in their study which analyze work engagement after Covid-19 found that marital status had a relationship with work engagement. Therefore, it has been seen that, post pandemic studies regarding work engagement investigates marital status as a significant element.

Conflicting results in the past research and the changes brought about by the COVID-19 pandemic in the business world highlight the importance of re-examining the relationship between marital status and work engagement among flexible workers in the post pandemic period. In this context, it is assumed that there is a relationship between marital status and work engagement.

H2a: There is a significant relationship between marital status and dedication among flexible workers

H2b: There is a significant relationship between marital status and absorption among flexible workers

H2c: There is a significant relationship between marital status and vigor among flexible workers

## The relationship between job experience and work engagement

Some scholars argue that there is an increase in work engagement as employees become more experienced [39, 40]. However, Mayer and Schoorman [107] observed the opposite indicating that there is a statistically significant negative relationship between job experience and work engagement. According to a study conducted by Faskhodi and Siyyari [41], among teachers work engagement increases as years of experience increase. In the study of Shukla et al., [38], it was seen that there was a significant difference in levels of work engagement depending on job experience. According to the study of Sharma et al., [102], it was found that there is a significant positive relationship between work engagement and age, education level and job experience. In the study of Bamford et al., [108] the number of years of nursing job experience was

found to be significantly and positively related to work engagement [108]. Poulsen et al., [35] showed that there is a positive relationship between job experience and work engagement. In the study of Bell and Barkhuizen [109], it was seen that there was no difference in work engagement in groupings according to gender, organizational level, number of years in the position and years of job experience. As can be seen, there are different results in the literature on the relationship between job experience and work engagement. This indicates that more research is needed on this subject. As has been discovered many times in the past literature, it is assumed in this study that there is a significant relationship between job experience and work engagement. In this context, the hypotheses developed with support from previous studies are as follows:

H3a: There is a significant relationship between job experience and dedication among flexible workers

H3b: There is a significant relationship between job experience and absorption among flexible workers

H3c: There is a significant relationship between job experience and vigor among flexible workers.

The proposed research model is depicted below in Fig 1.

## Materials and methods

Quantitative research method was used in this study. Web-based survey method was employed to collect data. Data were collected from January 1, 2021 to March 31, 2021. Web-based questionnaires were distributed to the flexible working participants in Istanbul, Turkey. The questions in the survey were in Turkish. Both scales used are 5-dimensional Likert scales. Expressions on the Likert scale range from 1-Strongly Disagree and 5-Strongly Agree. Initially, exploratory factor analysis (EFA) was applied to purify the data. Subsequently, confirmatory factor analyses (CFA) was performed to determine the convergent validity. Structural equation modelling method was used for CFA analysis. Then, Cronbach's alpha values were calculated to determine the reliability of the scales. Finally, T-test and ANOVA test were performed to test the hypotheses.

### Measures and sampling

To measure work engagement, the sub-dimensions measured are as follows: Vigor, Dedication and Absorption. The 17-item Utrecht Work Engagement Scale was used to measure the aforementioned constructs [71, 110]. Six of the items in the scale measure vigor, five items measure dedication and six items measure absorption. All the statements in the questionnaire were rated on a 5-point Likert scale where participants indicated whether they agreed with each statement (1 = Strongly Disagree, 5 = Strongly Disagree). 250 questionnaires were distributed and 199 valid questionnaires were collected from flexible workers from prominent companies in Turkey. All the participants were flexible workers. Convenience sampling method with voluntary response was utilized. The informed consent of participants was acquired and the research was conducted according to the ethical guidelines enunciated in the Declaration of Helsinki. The participants were ensured that their answers remain anonymous. Participants were classified into 3 different groups in terms of their experience. The first group consists of 51 people with up to 5 years of experience, the second group consists of 58 people with 5 to 10 years of experience, and the third group consists of 90 people with more than 10 years of

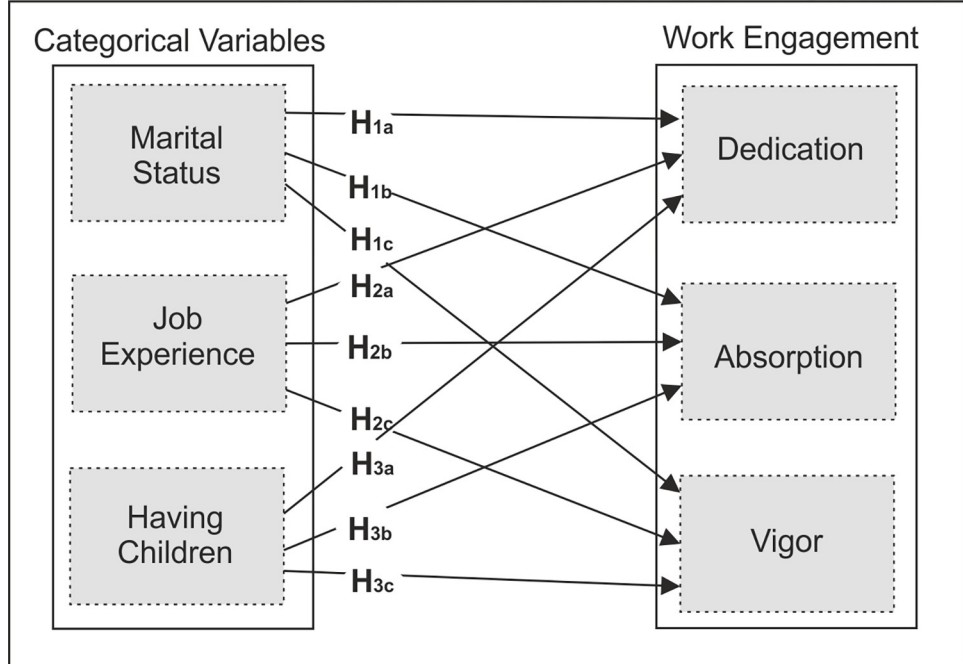

**Fig 1. Research model.**

experience. While 128 of the participants do not have children, 71 of them have children. % 56,3 of employees (112 of participants) are married and %43,7 of them are (87 of participants) single.

## Construct validity and reliability

First, principle component analysis was applied to purify the data and make the data ready for confirmatory factor analysis (CFA) [111]. Principle component analysis was used as exploratory. As a result, questions 4, 5, 6, 8, 11 and 12 were removed from the work engagement scale. 3 questions were removed from the vigor scale and 3 questions from the absorption scale. 3 questions were removed from the vigor scale and 3 questions from the absorption scale. Items "When I get up in the morning I feel like going to work.", "I can continue working for very long periods of at a time", "At my work I always persevere, even when things do not go well", I get carried away when I am working", "Time flies when I am working" and "I feel happy when I am working intensely" were removed. 11 items remained after exploratory factor analysis (EFA). Then convergent validity was determined by applying CFA. Fit indices values of the CFA was found satisfactory (i.e. $\chi2/DF = 1.632$, CFI = 0.983, IFI = 0.983, RMSEA = 0.056) [112]. Table 1 demonstrates the results of exploratory factor analysis and Table 2 shows the factor loads in CFA Results. As a result of the Cronbach's Alpha reliability test performed after the factor analysis, the Cronbach's Alpha reliability coefficient was 0.888 for vigor scale, 0.919 for dedication scale and 0,7 for absorption scale. The three dimensions and factor load in the work engagement scale are shown in Table 1.

Normal distribution assumption that is required for parametric statistical tests can be evaluated utilizing the values of skewness and kurtosis [113]. Although in social sciences it is mostly impossible to get a perfectly normal distribution, there are suggestions regarding the acceptable values of skewness and kurtosis. Tabachnik and Fidell [113] states that the interval of skewness and kurtosis values between -1,5 and +1,5 can be classified as normally distributed

**Table 1. Exploratory factor analysis and reliability test results.**

| Items | Factor loads | | | Cronbach Alpha |
|---|---|---|---|---|
| I am proud on the work that I do. (Dedication3) | 0,887 | | | 0,919 |
| My job inspires me. (Dedication4) | 0,861 | | | |
| I am enthusiastic about my job. (Dedication5) | 0,851 | | | |
| To me, my job is challenging. (Dedication2) | 0,816 | | | |
| I find the work that I do full of meaning and purpose. (Dedication2) | 0,793 | | | |
| At my job I feel strong and vigorous. (Vigor2) | | 0,920 | | 0,888 |
| At my work, I feel bursting with energy. (Vigor1) | | 0,903 | | |
| At my job, I am very resilient, mentally. (Vigor3) | | 0,810 | | |
| When I am working, I forget everything else around me. (Absorption3) | | | 0,841 | 0,700 |
| It is difficult to detach myself from my job. (Absorption4) | | | 0,758 | |
| I am immersed in my work. (Absorption1) | | | 0,734 | |

Extraction Method: Principal Component Analysis.

Rotation Method: Varimax with Kaiser Normalization.

Rotation converged in 5 iterations.

data. According to George and Mallery [114] a kurtosis value between +1 and -1 is considered exceptional for most psychometric goals. Table 3 presents the skewness and kurtosis values. In the light of Tabachnik and Fidell's and George and Mallery's [114] suggestions, it is observed that our data conforms to the normal distribution assumption [113].

## Results

T-test was conducted to test the relationship between work engagement and marital status. The decisions are taken at 0.05 significance level. According to the results, there is a difference in absorption, which is the sub-dimension of work engagement, according to marital status. Cleary, the other two sub-dimensions of work engagement namely dedication and vigor do not have a significant relevance with marital status. Accordingly, Hypotheses H2a ve H2c are not supported. Table 4 presents group statistics, Levene's test results and the significance values for T-test for equality of means concerning all of the sub-dimensions of work engagement.

**Table 2. Confirmatory factor analysis results.**

| Variables | Items | Standardized Factor Loads | Unstandardized Factor Loads |
|---|---|---|---|
| Dedication (DCT) | DCT05 | 0.866 | 1 |
| | DCT04 | 0.805 | 1.043 |
| | DCT03 | 0.894 | 1.040 |
| | DCT02 | 0.759 | 0.967 |
| | DCT01 | 0.821 | 0.972 |
| Vigor (VGR) | VGR03 | 0.577 | 1 |
| | VGR02 | 0.734 | 1.633 |
| | VGR01 | 0.705 | 1.586 |
| Absorption (ASP) | ASP04 | 0.676 | 1 |
| | ASP02 | 0.965 | 1.159 |
| | ASP01 | 0.928 | 0.922 |

a. p<0.01 for all items

**Table 3. Skewness and kurtosis values for work engagement.**

| Variable | Skewness | Kurtosis |
|---|---|---|
| Dedication | -0,887 | 0,766 |
| Vigor | -0,961 | 0,872 |
| Absorption | -0,072 | -0,335 |

The t-test results are presented in Table 4. According to this result, it is seen that the level of absorption of the employees change depending on whether they are married or not. Thus, Hypothesis 2b is accepted.

Table 5 shows ANOVA results which carried out at 0.05 significance level. Table 5 demonstrates that there is a significant relationship between job experience and work engagement. Therefore, Hypotheses 3a, 3b and 3c are supported. It is seen that those with 5 to 10 years of job experience have the highest average in terms of absorption. Looking at the averages, it is seen that more experienced employees are more dedicated and more vigorous. It is seen that the two groups which are more experienced, have higher absorption than the group with less experience.

T-test was also conducted to explore the relationship between having children and work engagement. Results of this analysis is given in Table 6. According to the results of this test, it is seen that the work engagement does not differ between those who have children and those who do not. Based on this result, it is seen that work engagement and having children are not related to each other.

## Discussion and conclusion

In this study, it has been investigated whether work engagement will change according to marital status, job experience and whether or not having children among flexible workers in the post COVID-19 era. The analysis results show that that there is a difference in absorption from the sub-dimensions of work engagement, depending on whether employees are married or not. In this context, it is understood that marital status among flexible workers after the pandemic is a significant factor in absorption of work. Therefore, it is understood that managers should consider marital status as an important factor while trying to increase the work engagement of the employees. It can be interpreted that married people are more concentrated on their work because their job experience increases in direct proportion to the possibility of being older than singles and because they stay at the place they work for a long time. In addition, in parallel with previous studies, it is seen that there is a difference in work engagement according to job experience [35, 102, 108]. It is seen that as the job experience increases, so

**Table 4. T-test results on the relationship between marital status and work engagement.**

| Group Statistics | | | | Items | | Levene's Test for Equality of Variances | | T-test for Equality of Means | |
|---|---|---|---|---|---|---|---|---|---|
| Items | Marital Status | N | Mean | | | F | Sig. | t | Sig. (2-tailed) |
| Mean Dedication | Married | 112 | 4,0357 | Mean Dedication | Equal variances assumed | 1,012 | 0,316 | 0,456 | 0,649 |
| | Single | 87 | 3,9793 | | Equal variances not assumed | | | 0,447 | 0,656 |
| Mean Vigor | Married | 112 | 4,1250 | Mean Vigor | Equal variances assumed | 2,485 | 0,117 | 0,925 | 0,356 |
| | Single | 87 | 4,0115 | | Equal variances not assumed | | | 0,948 | 0,345 |
| Mean Absorption | Married | 112 | 3,5625 | Mean Absorption | Equal variances assumed | 0,084 | 0,772 | 2,21 | 0,028 |
| | Single | 87 | 3,2835 | | Equal variances not assumed | | | 2,20 | 0,029 |

**Table 5. ANOVA test results on the relationship between job experience and work engagement.**

| ANOVA | | | | | | Test of Homogeneity of Variances | | | |
|---|---|---|---|---|---|---|---|---|---|
| Items | Groups | N | Mean | Sig. | | Items | | Levene Statistic | Sig. |
| Mean Vigor | 1 | 51 | 3,8889 | Between Groups | 0,061 | Mean Vigor | Based on Mean | 1,015 | 0,364 |
| | 2 | 58 | 4,0057 | | | | | 0,976 | 0,379 |
| | 3 | 90 | 4,2259 | | | | | 0,976 | 0,379 |
| Mean Dedication | 1 | 51 | 3,9843 | Between Groups | 0,781 | Mean Dedication | Based on Mean | 0,483 | 0,618 |
| | 2 | 58 | 3,9621 | | | | | 0,517 | 0,597 |
| | 3 | 90 | 4,0578 | | | | | 0,517 | 0,597 |
| Mean Absorption | 1 | 51 | 3,366 | Between Groups | 0,676 | Mean Absorption | Based on Mean | 1,548 | 0,215 |
| | 2 | 58 | 3,5172 | | | | | 1,47 | 0,232 |
| | 3 | 90 | 3,4333 | | | | | 1,47 | 0,232 |

Group 1: 0–5 years

Group 2: 5–10 years

Group 3: +10 years

does the work engagement of the people. This result is also supported by the previous studies [35, 102, 108]. Based on this result, it can be interpreted that more experienced and older people can be more vigorous towards their work due to their older age. It can be interpreted that the lack of vigor of those who have just started work and those with little job experience may be due to the fact that these people are open to new job opportunities and want to try different workplaces. Furthermore, the results surprisingly show that whether or not to have children is not associated with work engagement. This may be due to the older age of the children of the employees in the sample. At the same time, these findings may indicate that employees who spend more time with their children and for their personal lives thanks to their flexible working options are satisfied with this situation. The reason why a decrease has not been observed in the level of work engagement might be originated from the positive emotions rooted in spending more time with their children in a period of fear and stress.

Previously, Othman and Nasurdin [105] and Aboshaiqah et al., [37] showed that married employees were more engaged than their single counterparts. Similarly, this research partially supported previous studies by demonstrating the relationship of marital status with only one of sub-dimensions of work engagement namely absorption. The relationship between job experience and work engagement has also been discovered by various researchers [39, 40]. Likewise, present study also revealed that there is a significant relationship between job experience and work engagement among flexible workers in the post-pandemic period. Similarly

**Table 6. T-test results on the relationship of having a child and work engagement.**

| Group Statistics | | | | Items | | Levene's Test for Equality of Variances | | t-test for Equality of Means | |
|---|---|---|---|---|---|---|---|---|---|
| Items | | N | Mean | | | F | Sig. | t | Sig. (2-tailed) |
| Mean Dedication | 1 = without children | 128 | 3,9375 | Mean Dedication | Equal variances assumed | 0,056 | 0,813 | -1,62 | 0,107 |
| | 2 = with children | 71 | 4,1437 | | Equal variances not assumed | | | -1,675 | 0,096 |
| Mean Vigor | 1 = without children | 128 | 4,0026 | Mean Vigor | Equal variances assumed | 0,263 | 0,609 | -1,612 | 0,108 |
| | 2 = with children | 71 | 4,2066 | | Equal variances not assumed | | | -1,623 | 0,107 |
| Mean Absorption | 1 = without children | 128 | 3,4167 | Mean Absorption | Equal variances assumed | 0,07 | 0,932 | -0,506 | 0,613 |
| | 2 = with children | 71 | 3,4836 | | Equal variances not assumed | | | -0,511 | 0,610 |

with our results, Dumas and Perry-Smith [115] found that single, childless workers reported lower absorption than workers with other family structures [115]. According to their research, expecting household responsabilities after work strengthens work mindset, thereby rendering employees more involved in work. So, on the basis of this finding it can be deduced that having a spouse or children can influence work absorption [115]. Moreover, Zhang et al., [65] in their investigation concerning the effect of tremendous stress and excessive workload endured by nurses on work engagement after the Covid-19 pandemic, revealed that marital status is one of the main influencing factors of nurses' work engagement which signifies that married nurses showed higher work engagement. Similarly, Chan et al., [36] demonstrated that work engagement is related to being married. According to Balay-odao et al.,'s [106] research, marital status influences work engagement of millenial Saudi clinical nurses. It is discovered that married millenial Saudi nurses showed better dedication than single nurses. In addition, the reason for the high work engagement among married people is deemed to be originated by accountability, high involvement in work and emotional stability of married people and the extra support that they get from their families [106]. In a similar study conducted after the onset of the Covid-19 pandemic, considerably higher work engagement was noticed in people who were married or living with a partner, with children under 16 years of age, and with a very good perception of health in the last 14 days [32]. As a result, marital status and having children were found to be important factors in flexible workers after Covid-19. In a nutshell, as supported by previous similar studies, our findings suggests that seemingly employees balance both their personal obligations and their work well thanks to their flexible working arrangements, and that domestic responsibilities such as childcare are not a factor that reduces their work engagement.

In the focus of gender equality, this research examines work engagement according to having children and marital status, with the assumption that women's domestic work arising from traditional gender roles will become more destructive after Covid-19. In this context, present study discuss issues that are increasingly important such as family obligations and flexible working. It is identified that there are few studies similar to the current study after Covid-19. In this sense, it is anticipated that the current study will be a new and contributing study in the post pandemic era.

## Theoretical ımplications

This research supports social exchange and commitment-trust theories. It indirectly supports the social exchange theory, which proposes that flexible working arrangements will result in greater employee gratitude and greater motivation and loyalty to their work. The results show that even if people who are given the option of flexible working do have family obligations, their work engagement does not decrease. At the same time, the results also indirectly support the commitment-trust theory, which argues that policies that help reduce the problems of employees, provide work-life balance and that they will positively affect the loyalty and trust relations between the employee-employer and organizational performance.

The existing studies show that in the aftermath of the COVID-19 pandemic, women are overwhelmed by their increased domestic obligations as well as their workloads. In this context, the need to re-examine the variables of work engagement, marital status, job experience and having children, which were examined in the previous studies and which are related, emerged under the changing social and economic conditions in the post pandemic period [34, 39, 40, 65, 106]. In this context, this study contributes to filling this gap in the literature. Contrary to expectations, it is seen that the engagement of flexible workers does not decrease despite the additional obligations related to their family and children.

## Managerial implications

In addition to its contributions to theory, this research is also expected to be an important guide for managers in practice. Findings showing that as job experience increases, work engagement increases, managers can use flexible working arrangements to increase the work engagement of employees with more experience. It is understood that different mechanisms should be used to increase the level of work engagement of employees who have different obligations in their personal lives. This study reminds that managers should consider and take into account the roles and duties of the employees in their personal lives as well as their roles in business life. At the same time, this research reflects the situation of employees regarding the working order after the COVID-19 outbreak and informs managers about it. In addition, it is anticipated that flexible working has gained importance in the post COVID-19 period and will now be a tool that will be used more by managers to improve work-life balance in future research.

The findings of this research contribute to the scanty literature in post COVID-19 era by revealing the levels of work engagement according to the different demographic characteristics of flexible employees. Thus, it is anticipated that the findings of this study will indirectly contribute to the solution of new problems in employee behavior and work-life balance in the new post-pandemic period. The authors further accentuate this existing situation and point out to the fact that what may possible happen if return-to the-office policies are fully enforced.

## Limitations and future directions

This research has some limitations. First, the study was conducted in Turkey, and there-fore some of the results may have been based on the characteristics of Turkish culture. For this reason, it is important to carry out this research in different countries in the future and to make a cross-cultural comparison. In addition, the application of quantitative methods is another limitation of the study. Therefore, future research can enrich the literature in this field by applying qualitative research methods on flexible working and its consequences during and after COVID-19. In addition, future studies focusing on the impact of flexible working on leadership and teamwork during the COVID-19 era are encouraged. Considering the important relationship between work engagement and performance with the characteristics of colleagues and leaders, the impact of new ways of working on leadership and teams after the COVID-19 outbreak becomes important. Furthermore, the age of the children of the employees should be included as an important variable in future studies. In addition, the increasing burden of housework and childcare and the decrease in their participation in employment after the COVID-19 pandemic are among the most important problems in the post pandemic era [7]. Moreover, women seem to be 24% more likely than men to lose their jobs completely due to the COVID pandemic, and one year after the pandemic female employment dropped by %5,6 [116, 117]. In this context, future researchers are encouraged to work towards ensuring equality between men and women as well as solving these problems.

It is seen that new problems and new gaps continue to occur in the business and organ-izational behavior literature, which has undergone a great transformation after the epi-demic. In this context, it is understood that there is a need for similar studies in this field in the literature.

## Supporting information

**S1 Fig. Research model.**
(TIF)

**S1 Table. Exploratory factor analysis and reliability test results.**
(TIF)

**S2 Table. Confirmatory factor analysis results.**
(TIF)

**S3 Table. Skewness and kurtosis values for work engagement.**
(TIF)

**S4 Table. T-test results on the relationship between marital status and work engagement.**
(TIF)

**S5 Table. ANOVA test results on the relationship between job experience and work engagement.**
(TIF)

**S6 Table. T-test results on the relationship of having a child and work engagement.**
(TIF)

**S1 Appendix.**
(DOCX)

## Author Contributions

**Conceptualization:** Murat Çemberci, Mustafa Emre Civelek, Adnan Veysel Ertemel, Perlin Naz Cömert.

**Data curation:** Mustafa Emre Civelek, Perlin Naz Cömert.

**Formal analysis:** Mustafa Emre Civelek, Perlin Naz Cömert.

**Investigation:** Murat Çemberci.

**Methodology:** Mustafa Emre Civelek.

**Project administration:** Murat Çemberci, Adnan Veysel Ertemel, Perlin Naz Cömert.

**Resources:** Murat Çemberci, Mustafa Emre Civelek, Adnan Veysel Ertemel, Perlin Naz Cömert.

**Software:** Mustafa Emre Civelek.

**Supervision:** Murat Çemberci, Adnan Veysel Ertemel.

**Validation:** Mustafa Emre Civelek, Perlin Naz Cömert.

**Visualization:** Mustafa Emre Civelek.

**Writing – original draft:** Murat Çemberci, Adnan Veysel Ertemel, Perlin Naz Cömert.

**Writing – review & editing:** Adnan Veysel Ertemel, Perlin Naz Cömert.

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
