## [Decision Letter · Decision Letter 0]

11 Aug 2022

PONE-D-22-08087The Relationship of Work Engagement with Job Experience, Marital Status and Having Children Among Flexible Workers After the Covid-19 PandemicPLOS ONE

Dear Dr. Adnan Veysel Ertemel,

Thank you for submitting your manuscript to PLOS ONE. After careful consideration, we feel that it has merit but does not fully meet PLOS ONE’s publication criteria as it currently stands. Therefore, we invite you to submit a revised version of the manuscript that addresses the points raised during the review process.

We look forward to receiving your revised manuscript.

Kind regards,

Rogis Baker, Ph.D

Academic Editor

PLOS ONE

Journal Requirements:

2. Please provide additional details regarding participant consent. In the Methods section, please ensure that you have specified (1) whether consent was informed and (2) what type you obtained (for instance, written or verbal). If your study included minors, state whether you obtained consent from parents or guardians. If the need for consent was waived by the ethics committee, please include this information.

3. We note that you have referenced ( Ivanauskaite A.et al. [15]) which has currently not yet been accepted for publication. Please remove this from your References and amend this to state in the body of your manuscript: ( Ivanauskaite A.et al. [Unpublished]”) as detailed online in our guide for authors

Reviewers' comments:

Reviewer's Responses to Questions

**Comments to the Author**

1. Is the manuscript technically sound, and do the data support the conclusions?

Reviewer #1: Yes

Reviewer #2: Partly

Reviewer #3: Yes

Reviewer #4: Partly

2. Has the statistical analysis been performed appropriately and rigorously? 

Reviewer #1: Yes

Reviewer #2: No

Reviewer #3: Yes

Reviewer #4: No

3. Have the authors made all data underlying the findings in their manuscript fully available?

Reviewer #1: Yes

Reviewer #2: Yes

Reviewer #3: Yes

Reviewer #4: Yes

4. Is the manuscript presented in an intelligible fashion and written in standard English?

Reviewer #1: Yes

Reviewer #2: Yes

Reviewer #3: Yes

Reviewer #4: Yes

5. Review Comments to the Author

Reviewer #1: The paper analyses an interesting topic which given the nature and actuality of the subject (work life balance), should be of interest to the readership.

I have the following proposed amendments:

(1) For the Introduction part it would be good to have some statistics on the number of Covid-19 cases, deaths. Please add some data!

(2) The Conceptual Background chapter is too mosaic, the Author should add more literature and not only refer to the two topics (Flexible Work Options and Work Engagement).

(3) I propose to reconsider and reformulate the hypotheses. The Authors have formulated too many hypotheses.

(4) The Authors wrote in line 342. "Quantitative research method was used in this study." We have no more information about this research. We can read only in the abstratc some details. It would be necessary to explain the methodology, when the survey was conducted and with whom!

(5) Because of the reformulation of the hypotheses, the summary also requires rethinking.

Reviewer #2: GENERAL COMMENTS FROM REVIEWER

MANUSCRIPT NUMBER: PHONE-D-22-08087

I. The abstract is not clear, re-write it considering context, aim(s), method, results and implications.

II. Novelty and contribution should be clarified.

III. Motivation behind the work should be forcefully discussed.

IV. More discussions are essential.

V. Comparison analysis with more articles should be discussed.

Reviewer #3: Some review comments to the author:

1. Abstract: research method should be included in this part. Besides, the results of marital status with other factors of work engagement, except absorption are not summarized.

2. Reference No. 15 is an unpublished master thesis, which should not be cited.

3. Conceptual background: are there any other studies about flexible work options in the context of post Covid-19? Comparison is needed in this case.

4. Development of hypotheses: this part needs to incoporate a comparison of these relationships in this study with other studies having the same research context to show the development of hyphotheses.

5. Materials and Methods:

- How to determine the sample size for this study?

- How to make sure the representativeness of sample?

- What are the demographics of sample?

- Given human participation, whether ethics statement including ethical approval and informed consent is needed or not.

6. Results:

- What is the basis of dividing job experience into 3 groups as in the article?

- Line 376 and 388, the number of table is incorrectly stated.

7. Discussion and Conclusion: the findings of this study should be compared with other studies of the same context.

Reviewer #4: Comments to the Author:

Thank you for the opportunity to review this paper. It is very well written, and it addresses an essential topic. The authors have invested a lot of knowledge and attention into the matter. These results will contribute to the growing expertise in the field. However, some parts of the study should be revised. The fact that the confirmatory analysis (CFA) was not made is critical. CFA should be done.

General comments:

1. The methodology is not described in the abstract.

2. Methods: Authors did an explorative factor analysis (EFA) to prove the scale's validity. That is good, but that is just the first step. The second step is confirmatory analysis (CFA). However, the Authors should conduct EFA and CFA to test scale validity. At this level, the CFA can determine the average extracted variance and discriminant validity using the Fornell-Larcker criterion.

3. How large was the sample? You are writing about 199 valid questionnaires, but there is no information about the sample and response rate. That should be added to the article.

4. In which country and in which language did you make your data collection? This information is missed. Please add as supplement materials the questionnaire in the language in which the survey was conducted.

5. How did you collect the data? Did you use a Paper- or Web-Based Questionnaire? This information is missed.

6. Results – Please describe in more detail the results from tables 1,2,3, and 4 in the main text. Write the main results from the hypothesis here. You have written just a sentence that is not enough.

7. Tables - Please write tables in such a way that it is easy to read. Now everything is shifted, which doesn't look good and reads badly.

8. Discussion and Conclusion – You should start the discussion with your findings, not with findings from other studies. However, this section should be structured in the following sections:

a. Overview of findings

- What were the main findings?

b. Comparisons with other studies

c. Strengths and limitations of research

d. Implications of findings

- E.g., What items were not important -why do you think this was?

Future directions for research

Conclusion

- E.g., What do you think are the key take-home messages?

Specific comments:

1. Introduction, paragraph 1, lines 26-27 - A source is missing here.

2. Introduction, lines 58-59 – Where was this study conducted? Is it related to your work? Source (9).

3. Introduction, line 77 – Here, you have a hyphen too many.

4. Introduction, line 125 – Here, you have a hyphen too many.

5. Introduction, line 128 – Here, you have a hyphen too many.

6. Introduction, line 132 – Here, you have a hyphen too many.

7. Conceptual Background, line 141 – Here, you have a hyphen too many.

8. Conceptual Background, line 194 – Here, you have a hyphen too many.

9. Conceptual Background, line 216 – Here, you have a hyphen too many.

10. Conceptual Background, line 223, 227,236 – Here, you have a hyphen too many.

11. Development of Hypotheses lines 251,263,292 – Here, you have a hyphen too many.

12. Construct Validity and Reliability – Please write which items were excluded in the table or in the text.

13. Table 1. Factor and Reliability Analysis Results - Please write the names of the items in the table. This way, the reader can get a feel for the scale. Please add cross and side loadings for each item.

14. Results, lines 378-380 – This sentence should be moved to the discussion. “It can be interpreted that married people are more concentrated on their work because their job experience increases in direct proportion to the possibility of being older than singles and because they stay at the place they work for a long time.”

15. Results - This sentence should be moved to the methods. Additionally, it would be good if you wrote how many participants were in each group. “Participants were classified into 3 different groups in terms of their experience. The first group consists of those with up to 5 years of experience, the second group consists of those with 5 to 10 years of experience, and the third group consists of those with more than 10 years of experience.”

6. PLOS authors have the option to publish the peer review history of their article (what does this mean?). If published, this will include your full peer review and any attached files.

Reviewer #1: **Yes: **Katalin Liptak

Reviewer #2: No

Reviewer #3: No

Reviewer #4: No

---

## [Author Response · Author response to Decision Letter 0]

6 Oct 2022

Dear Dr. Rogis Baker, 

Thank you for giving us the opportunity to submit a revised draft of the manuscript “The Relationship of Work Engagement with Job Experience, Marital Status and Having Children Among Flexible Workers After the Covid-19 Pandemic” for publication in the Journal Plos One. We appreciate the time and effort that you and the reviewers dedicated to providing feedback on our manuscript and are grateful for the insightful comments on and valuable improvements to our paper. We have incorporated most of the suggestions made by the reviewers. Those changes are highlighted within the manuscript. Please see below, for a point-by-point response to the reviewers’ comments and concerns. All line numbers refer to the revised manuscript file.

REVIEWER 1 

1.For the Introduction part it would be good to have some statistics on the number of Covid-19 cases, deaths. Please add some data!

As suggested by reviewer, we added statistics related to Covid-19 in line 35. 

2.The Conceptual Background chapter is too mosaic, the Author should add more literature and not only refer to the two topics (Flexible Work Options and Work Engagement).

As suggested by reviewer, we added more literature in line 229-233 about recent studies examining flexible work and gender equality which is also related to our research. However, we did not examine gender equality topic in depth in order not to deviate from our main research topic. Furthermorei between lines 273-293 literature review related to marital status, having children and job experience has been added. 

3.I propose to reconsider and reformulate the hypotheses. The Authors have formulated too many hypotheses.

While we appreciate the reviewer’s feedback, we respectfully disagree. We think that the number of hypotheses is appropriate. However we did a revision in formulation of hypotheses. 

4.The Authors wrote in line 342. “Quantitative research method was used in this study.” We have no more information about this research. We can read only in the abstratc some details. It would be necessary to explain the methodology, when the survey was conducted and with whom!

We thank the reviewer for valuable feedbacks. We added details about research methodology. 

5.Because of the reformulation of the hypotheses, the summary also requires rethinking.

We made the relevant modifications in the abstract. 

REVIEWER 2

1. The abstract is not clear, re-write it considering context, aim(s), method, results and implications.

We thank the reviewer for pointing this out. The revised abstract can be found in the revised manuscript. 

2. Novelty and contribution should be clarified.

We added more discussion related to the novelty and contribution of the research in between lines 554-559. 

3. Motivation behind the work should be forcefully discussed.

Motivation behind the work is discussed in lines 128-133 and 107-110. 

4. More discussions are essential.

Thank you for this suggestion. We added more discussions in lines 523-559, between 569-576 and between 606-612. 

5.Comparison analysis with more articles should be discussed.

Between the lines 526-549 comparisons with more past studies is made. 

REVIEWER 3

1.Abstract: research method should be included in this part. Besides, the results of marital status with other factors of work engagement, except absorption are not summarized.

Thank you for your valuable feedback. The relevant changes in the abstract have been made.

2. Reference No. 15 is an unpublished master thesis, which should not be cited.

As suggested by reviewer, this source is eliminated. 

3.Conceptual background: are there any other studies about flexible work options in the context of post Covid-19? Comparison is needed in this case.

We have added the suggested content to the manuscript in lines 229-234. 

4.Development of hypotheses: this part needs to incoporate a comparison of these relationships in this study with other studies having the same research context to show the development of hyphotheses.

Newly added parts between lines 299-308, 374-377 and between lines 347-352 reflect a comparison and a concluding remark of previous studies.

5. Materials and Methods:

- How to determine the sample size for this study?

- How to make sure the representativeness of sample?

- What are the demographics of sample?

- Given human participation, whether ethics statement including ethical approval and informed consent is needed or not. 

Thank you for valuable feedbacks, we answered reviewer’s questions in lines 428-438. 

6. Results:

- What is the basis of dividing job experience into 3 groups as in the article?

- Line 376 and 388, the number of table is incorrectly stated.

We gave detailed explanations about job experience groups in lines 434-438. Mistakes regarding the number of tables is corrected in the revised manuscript. 

7. Discussion and Conclusion: the findings of this study should be compared with other studies of the same context.

As suggested by reviewer, the relevant comparisons have been added in lines 524-551. 

REVIEWER 4: 

1.The methodology is not described in the abstract.

The description of methodology is added in the abstract. 

2.Methods: Authors did an explorative factor analysis (EFA) to prove the scale's validity. That is good, but that is just the first step. The second step is confirmatory analysis (CFA). However, the Authors should conduct EFA and CFA to test scale validity. At this level, the CFA can determine the average extracted variance and discriminant validity using the Fornell-Larcker criterion.

Thank you for reviewer’s valuable feedback. The CFA is conducted and the relevant details is explained in lines 417-421 and 440-447. CFA results is presented in Table 2 in line 455. 

3. How large was the sample? You are writing about 199 valid questionnaires, but there is no information about the sample and response rate. That should be added to the article.

The necessary information is added between lines 429- 439. 

4. In which country and in which language did you make your data collection? This information is missed. Please add as supplement materials the questionnaire in the language in which the survey was conducted. 

Relevant details about data collection is described in lines 414-421 The questionnaire is added in Appendix. 

5. How did you collect the data? Did you use a Paper- or Web-Based Questionnaire? This information is missed.

This information added to the manuscript in line 413. 

6. Results – Please describe in more detail the results from tables 1,2,3, and 4 in the main text. Write the main results from the hypothesis here. You have written just a sentence that is not enough.

Results are described in more detail in lines 471-475. In lines 473, 481, 490 we explained which hypotheses are accepted and which are not accepted. In line 488 and 470 we explained the level of significance. 

7. Tables - Please write tables in such a way that it is easy to read. Now everything is shifted, which doesn't look good and reads badly.

We agree with the reviewer’s assessment. We fixed the tables accordingly. 

8. Discussion and Conclusion – You should start the discussion with your findings, not with findings from other studies. However, this section should be structured in the following sections:

a. Overview of findings

- What were the main findings?

b. Comparisons with other studies

c. Strengths and limitations of research

d. Implications of findings

- E.g., What items were not important -why do you think this was?

Future directions for research

Conclusion

- E.g., What do you think are the key take-home messages?

We made the necessary changes about the structure of discussion and conclusion part. We restructured the discussion part and we added Theoretical Implications, Managerial Implications and Limitations and Future Directions sections after the discussion and conclusion. We structured the order of the sections as suggested by the reviewer. Also, between the lines 524-551 we added past similar studies and compared them with our results. 

Specific comments:

1. Introduction, paragraph 1, lines 26-27 - A source is missing here.

The mentioned phrase is removed. 

2. Introduction, lines 58-59 – Where was this study conducted? Is it related to your work? Source (9).

The relevant information about the aforementioned study is added in the manuscript. 

3. Introduction, line 77 – Here, you have a hyphen too many.

4. Introduction, line 125 – Here, you have a hyphen too many.

5. Introduction, line 128 – Here, you have a hyphen too many.

6. Introduction, line 132 – Here, you have a hyphen too many.

7. Conceptual Background, line 141 – Here, you have a hyphen too many.

8. Conceptual Background, line 194 – Here, you have a hyphen too many.

9. Conceptual Background, line 216 – Here, you have a hyphen too many.

10. Conceptual Background, line 223, 227,236 – Here, you have a hyphen too many.

11. Development of Hypotheses lines 251,263,292 – Here, you have a hyphen too many.

The hypens were removed. 

12. Construct Validity and Reliability – Please write which items were excluded in the table or in the text.

Which items were excluded is explained between lines 442-447. 

13. Table 1. Factor and Reliability Analysis Results - Please write the names of the items in the table. This way, the reader can get a feel for the scale. Please add cross and side loadings for each item.

The names of the items is written in the table 1 line 454. 

14. Results, lines 378-380 – This sentence should be moved to the discussion. “It can be interpreted that married people are more concentrated on their work because their job experience increases in direct proportion to the possibility of being older than singles and because they stay at the place they work for a long time.”

As suggested by reviewer this sentence is moved to the discussion section in line 510. 

15. Results - This sentence should be moved to the methods. Additionally, it would be good if you wrote how many participants were in each group. “Participants were classified into 3 different groups in terms of their experience. The first group consists of those with up to 5 years of experience, the second group consists of those with 5 to 10 years of experience, and the third group consists of those with more than 10 years of experience.”

As suggested by reviewer this sentence is moved to the measures and sampling section in line 434.

---

## [Editor Report · Decision Letter 1]

14 Oct 2022

The Relationship of Work Engagement with Job Experience, Marital Status and Having Children Among Flexible Workers After the Covid-19 Pandemic

PONE-D-22-08087R1

Dear Dr. Adnan Veysel Ertemel,

We’re pleased to inform you that your manuscript has been judged scientifically suitable for publication and will be formally accepted for publication once it meets all outstanding technical requirements.

Kind regards,

Rogis Baker, Ph.D

Academic Editor

PLOS ONE
---

## [Editor Report · Acceptance letter]

20 Oct 2022

PONE-D-22-08087R1 

The Relationship of Work Engagement with Job Experience, Marital Status and Having Children Among Flexible Workers After the Covid-19 Pandemic 

Dear Dr. Ertemel:

I'm pleased to inform you that your manuscript has been deemed suitable for publication in PLOS ONE. Congratulations! Your manuscript is now with our production department. 

Kind regards, 

on behalf of

Dr. Rogis Baker 

Academic Editor

PLOS ONE